# Peritoneal Flap Following Lymph Node Dissection in Robotic Radical Prostatectomy: A Novel “Bunching” Technique

**DOI:** 10.3390/cancers16081547

**Published:** 2024-04-18

**Authors:** Ahmed Gamal, Marcio Covas Moschovas, Abdel Rahman Jaber, Shady Saikali, Marco Sandri, Ela Patel, Evan Patel, Travis Rogers, Vipul Patel

**Affiliations:** 1AdventHealth Global Robotics Institute, Celebration, FL 34747, USAasjaber@hotmail.com (A.R.J.);; 2Urology Department, University of Central Florida (UCF), Orlando, FL 32816, USA; 3Big and Open Data Innovation Laboratory (BODaI-Lab) and Data Methods and Systems Statistical, 25123 Brescia, Italy; sandri.marco@gmail.com; 4Stanford University, Palo Alto, CA 94305, USA

**Keywords:** radical prostatectomy, robot-assisted, pelvic lymph node dissection, lymphocele, symptomatic lymphoceles, peritoneal flap

## Abstract

**Simple Summary:**

Symptomatic lymphocele can occur in 2–8% of patients following PLND in RARP, and the condition can add significant morbidities, such as fever, lower abdominal pain, deep venous thrombosis (DVT), and lower urinary tract symptoms. The literature has outlined diverse techniques for preventing the formation of lymphoceles, including ligature, clipping, or the mono/bipolar sealing of lymphatic vessels. Nevertheless, there exists ongoing controversy and debate regarding the efficacy of these varied approaches. Recent studies suggest that incorporating a peritoneal flap (PF) may reduce the risk of lymphoceles by enhancing the absorption of lymph fluid through the peritoneum. We described a novel technique for peritoneal flap creation that involves bunching the bladder peritoneum and suturing it to the periosteum of the pubic bone to secure it in place. Our technique has been shown to be effective in elimination of symptomatic lymphocele formation. The bunching technique is safe, feasible, does not add significant morbidities, and does not require a learning curve.

**Abstract:**

Background: Pelvic lymph node dissection (PLND) is recommended while performing robot-assisted radical prostatectomy (RARP) for patients with localized intermediate or high-risk prostate cancer. However, symptomatic lymphoceles can occur after surgery, adding significant morbidity to patients. Our objective is to describe a novel Peritoneal Bladder Flap Bunching technique (PBFB) to reduce the risk of clinically significant lymphoceles in patients undergoing RARP and PLND. Methods: We evaluated 2267 patients who underwent RARP with PLND, dividing them into two groups: Group 1, comprising 567 patients who had the peritoneal flap (PBFB), and Group 2, comprising 1700 patients without the flap; propensity score matching carried out at a 1:3 ratio. Variables analyzed included estimated blood loss (EBL), operative time, postoperative complications, lymphocele formation, and the development of symptomatic lymphocele. Results: The two groups exhibited similar preoperative characteristics after matching. There was no statistically significant difference in the occurrence of lymphoceles between the flap group and the non-flap group, with rates of 24% and 20.9%, respectively (*p* = 0.14). However, none of the patients in the flap group (0%) developed symptomatic lymphoceles, whereas 2.2% of patients in the non-flap group experienced symptomatic lymphoceles (*p* = 0.01). Conclusion: We have demonstrated a modified technique for a peritoneal flap (PBFB) with the initial elimination of postoperative symptomatic lymphoceles and promising short-term outcomes.

## 1. Introduction

Robot-assisted radical prostatectomy (RARP) has become a popular surgical approach for localized prostate cancer due to its favorable oncological and functional outcomes, as well as lower morbidity. In cases of intermediate- and high-risk prostate cancer, bilateral pelvic lymphadenectomy (PLND) is recommended as an adjunct to RARP [1,2]. However, the PLND extension and optimal template are still under debate regarding high-level evidence on this topic [3,4]. Despite its benefits, PLND can lead to surgical complications, with postoperative lymphocele formation being the most common. Most postoperative lymphoceles are clinically insignificant and found incidentally during follow-up with variable incidence across the literature, reaching up to 60% of cases [5,6]. However, a small percentage of patients, i.e., 2–8%, may experience symptomatic lymphoceles (SL), which can cause significant morbidity [5,7,8].

Various techniques have been described in the literature to prevent lymphocele formation. These techniques involve ligature, clipping, or the mono/bipolar sealing of lymphatic vessels. However, there is ongoing controversy and debate regarding the effectiveness of these different approaches [9,10]. Additionally, alternative approaches, including applying fibrin glue, FloSeal^®^, or TachoSil^®^ during surgery, have been explored, but their clinical significance in preventing lymphocele formation remains uncertain, as studies have not been able to show the significant superiority of these hemostatic patches [11,12,13].

Recent studies have proposed that a peritoneal flap (PF) might reduce the risk of asymptomatic and symptomatic lymphoceles by facilitating the absorption of lymph fluid by the peritoneum [14,15,16,17]. In our study, we describe a new technique for PFs (PBFB), and we evaluate the effectiveness of this technique in decreasing the rate of postoperative symptomatic lymphoceles following PLND in RARP.

## 2. Materials and Methods

### 2.1. Inclusion Criteria

We analyzed the data of patients from 2010 to 2023 who underwent RARP and PLND at our institution by a single surgeon (V.P). The database was collected prospectively and reviewed under an IRB-approved protocol. We stratified patients into two groups: 567 patients who had RARP and PLND with PF (PBFB) from November 2022 to March 2023 and 1700 who had RARP and PLND without the flap selected using propensity score (PS) matching. We excluded patients who underwent previous pelvic radiation.

Perioperative and postoperative variables were recorded and analyzed for significant differences in outcomes between groups. Variables analyzed were estimated blood loss (EBL), console time, hospitalization, postoperative complications (Clavien Dindo), and the development of lymphoceles (size, symptomatic lymphocele) [18]. Patients were monitored clinically in the postoperative setting and at catheter removal. A pelvic ultrasound was carried out for all patients at 6-week follow-up, and additional clinical follow-up was carried out at 3 months following surgery. If a lymphocele was found on the initial ultrasound, ultrasounds were then repeated every 3 months for the first year.

### 2.2. Endpoints

Our primary endpoint was to evaluate the effectiveness of Peritoneal Bladder Flap Bunching (PBFB) on lymphocele formation and SL development.

### 2.3. Propensity Score Matching

To reduce the biasing effect of potential confounders in our data, 567 patients who underwent RARP and PLND with the PBFB were matched with 1700 patients (1:3 ratio) from a cohort of 4017 who underwent RARP and PLND using the da Vinci surgical platform (Table 1). The propensity score (PS) was estimated using a multivariable logistic regression model considering the following variables: age, BMI, American Urological Association symptom score (AUASS), PSA levels, DM, and ISUP biopsy grading.

Matching was performed using the nearest-neighbor matching algorithm (caliper width was 0.25 of the standard deviation of the logit score) with a 1:3 ratio without replacement. The quality of matching was evaluated using the standardized mean difference, which is a measure of the degree of covariate imbalance [19]. Covariates with a standardized difference of <0.15 in the absolute value were considered satisfactorily balanced. In addition, we used the nonparametric Kolmogorov–Smirnov test to assess the equality of distributions for continuous variables and the Fisher’s exact test for categorical variables. Table 2 describes the whole cohort included before PS matching and Figure 1 illustrates the love plot built to describe the variables balance before and after matching [20].

### 2.4. Statistical Analysis

Preoperative and perioperative patient characteristics were summarized as the median and interquartile range (IQR) for continuous variables and absolute and relative frequencies for categorical variables. Comparisons of perioperative continuous variable distributions between the two study groups were investigated using the nonparametric Wilcoxon rank-sum test. For categorical variables, their distributions were compared using Fisher’s exact test.

Statistical analyses were conducted utilizing Stata 16 (StataCorp 2019, College Station, TX, USA) and R 4.3.1 (R Core Team 2021; R Foundation for Statistical Computing, Vienna, Austria). A *p*-value threshold of 0.05 was adopted as the criterion for statistical significance.

#### 2.4.1. Surgical Technique

All patients underwent our conventional transperitoneal RARP and nerve-sparing technique using six surgical ports [21,22,23]. The decision to perform PLND was based on patient risk stratification using the D’Amico score; high- and intermediate-risk patients routinely received PLND, while low-risk cases were assessed individually by the single performing surgeon [24]. The standard template for PLND boundaries includes the external iliac vein on the lateral side, the bladder on the medial side, and the obturator nerve posteriorly. The administration of 5000 units of subcutaneous heparin occurred immediately after anesthesia induction and was continued twice daily until discharge. The techniques for RARP and PLND used have been described previously in studies from our center [21,22,23,25].

#### 2.4.2. Peritoneal Bladder Flap Bunching Technique (PBFB)

From November 2022 onwards, all patients undergoing a PLND would undergo the creation of PF. In the intervention group, after vesicourethral anastomosis, a modified PF was created where the bladder peritoneum that was released to enter the retropubic space is bunched together and was attached to the midline pubic tubercle. Using a 2-0 Quill™ barbed suture, the first stitch in the peritoneal fold on the right side medial to the vase, Figure 2B, is followed by a stitch of the peritoneal fold on the left side, Figure 2C; this approximates the two edges of the peritoneum in the midline, and a running suture was passed through the bladder peritoneum from both sides in a bunching fashion, as seen in Figure 2E,F. In sequence, this suture was then passed through the periosteum of the pubic bone and fixed to the periosteum of the pubic bone to be held in place, as seen in Figure 2H. This technique leaves the lateral gutters of the pelvis open for lymphatic drainage and avoids the complete closure of the peritoneum and peri-vesical fat over the dissection bed, allowing proper lymphatic fluid drainage from the true pelvis into the abdomen while avoiding any disruption of the anastomosis. Some lymphatic fluid can still collect because the obturator fossa is dependent and closed; however, a large volume lymphatic fluid accumulation is prevented. Figure 2 illustrates the peritoneal bunching technique in patients with PLND.

## 3. Results

### 3.1. Patient Demography

Table 1 shows that, after 1:3 PS matching, the groups were satisfactorily balanced and exhibited similar preoperative characteristics, including age, PSA, BMI, ISUP biopsy grading, and AUA scores.

### 3.2. Intraoperative Parameters

Regarding intraoperative parameters (Table 3), the total console time was higher in the flap group. In the flap group, 232 patients (41.1%) had an operative time of ≥90 min, compared to 552 patients (32.7%) in the non-flap group (*p* < 0.001). The percentage of patients with an estimated blood loss (EBL) greater than 100 mL was 43% in the flap group, compared to 27.5% in the control group (*p* < 0.001). However, none of the patients in either group required blood transfusions.

No immediate intraoperative complications were observed in the two groups. The median duration required for the creation of a peritoneal flap was 3 min 15 s, with an interquartile range (IQR) spanning from 2 min 50 s to 3 min 25 s.

### 3.3. Postoperative Parameters and Complications

We found no statistically significant differences in complication rates between the groups, as shown in Table 3. Similarly, the difference in lymphocele formation between the PF group and the control group was not statistically significant (24% vs. 20.9%; *p* = 0.14). In the subgroup analysis, we observed that the flap group had a significantly lower incidence of 2.1% in large lymphoceles (>5 cm), compared to 11.1% in the non-flap group (*p* < 0.001). Notably, none of the patients in the flap group experienced symptomatic lymphoceles (0%), whereas 37 patients (2.2%) in the non-flap group developed symptomatic lymphoceles, all of them requiring percutaneous drainage (*p* = 0.01). This difference remains significant in a multivariable analysis, considering the four potential confounders of obesity, DM, pT, and total number of LNs.

## 4. Discussion

Pelvic lymph node dissection (PLND) is considered the most effective method for assessing lymph node metastases in patients with clinically localized prostate cancer. By improving cancer staging, PLND allows for the better risk assessment of cancer local status after RARP and helps identify patients who potentially benefit from additional adjuvant therapy [2,3]. However, the optimal lymph node template (extension of dissection) and its impacts on improving patient overall survival are still under debate [3,4].

In this context, PLND is not devoid of morbidity and possible complications, with lymphoceles being the most common. Many attempts to prevent lymphocele formation have been investigated and described; however, none of these techniques have proved efficacious in preventing lymphocele formation [9,10,11].

Stolzenburg et al. were one of the first to hypothesize that peritoneal fenestration allows free lymph drainage into the abdomen to be absorbed in extraperitoneal radical prostatectomies [17]. In the current popular transperitoneal approach, Lebies et al. introduced the peritoneal flap by securing the available peritoneum surrounding the posterior bladder to the lateral aspect of the bladder using multiple interrupted sutures. The author explained that lymphoceles commonly occur after PLND because bladder adhesion forms a barrier around the lymphadenectomy bed, preventing the lymphatic fluid from draining into the peritoneal cavity. In these cases, the peritoneal interposition flap (PIF) prevents the bladder from forming a scar over the lymphadenectomy bed and instead creates a window through which the lymphatic fluid can freely drain into the peritoneal cavity to be reabsorbed. This study revealed that the incidence of SL was significantly lower in patients who underwent the PIF process compared to those who did not receive a flap. Among the group of 77 patients who received a PIF, there were no cases of SL (0.0%), whereas in the group of 77 patients without a PIF, 11.6% (9 individuals) experienced SL. This difference in lymphocele formation between the two groups was statistically significant [15].

In subsequent studies, Stolzenburg et al. introduced a modification to the peritoneal flap to address the possible effect on micturition when fixing the flap to the bladder wall. In their technique, the “four-point peritoneal flap fixation” (4PPFF), the peritoneal flap is fixed at four points to the anterior and lateral pelvic sidewalls. The study revealed a decreased incidence of SL in the group of 4PPFF compared to the control group (1.0% versus 4.6%; *p* = 0.032) [14].

In our technique (PBFB), the peritoneal free flap on both sides is bunched together, creating one midline flap fixed to the pubic bone near the symphysis pubis, allowing open lateral gutters and good lymphatic drainage into the abdomen while keeping the lymphadenectomy bed and obturator fossa unobstructed. While this technique did not eliminate the occurrence of lymphoceles, there was a significant reduction (almost 9%) in lymphoceles formation measuring > 5 cm compared to the non-flap group (2.12% vs. 11.12%, respectively). Moreover, there was no incidence of SL in the flap group compared to patients in the non-flap group (*p* = 0.001, Fisher’s exact test). The reason that some lymphatic fluid collection persisted in the PBFB group is because the obturator fossa is dependent and often a closed space. However, in the PBFB group there is still drainage as the lateral gutters are open due to a lack of complete peritonealization. In patients without a flap, the peritoneum often quickly reperitonealizes in less than 48 h and closes the space to lymphatic drainage.

In a prospective multicenter randomized controlled trial (ProLy study) involving a total of 475 patients, the occurrence of lymphoceles was assessed in two groups: Group A (239 patients with PIF) and Group B (236 patients without PIF). PIFs were established by stitching the edges of the bladder peritoneum to the endopelvic fascia on the same side at two distinct locations. Lymphoceles were identified in 22% of patients in Group A and 33% in Group B, with a statistically significant difference (*p* = 0.008). Furthermore, there was a significant reduction in SL by almost 5% (3.3% vs. 8.1% in Group A and Group B, respectively) [26].

In the same context, Deutsch et al. conducted a meta-analysis of five retrospective studies, which showed a 77% reduction in SL formation (95% CI: 1–99%). The meta-analysis revealed that the incidence of SL was 1.3% (8 out of 604 patients) in the PIF group, compared to 5.7% (40 out of 704 patients) in the standard group (*p* < 0.001) [27]. Showing a summary of studies describing the potential benefits of this technique.

On the other hand, a recent prospective randomized study, named the PIANOFORTE trial, could not find any significant benefit of PIF. In their study, 232 patients were divided into two groups: 108 patients in the intervention group (peritoneal flap) and 124 patients in the control group (no flap). The peritoneal flap was generated following the completion of the vesicourethral anastomosis, in precise accordance with the description provided in the original study by Lebeis et al. [15]. The results indicated a lower incidence of lymphoceles after PIF (18% vs. 24%), but without statistical significance (*p* = 0.65). There was also no statistically significant difference in SL incidence between the two groups (8.3 vs. 9.7 *p* = 0.82). They emphasized that further research is needed to address the existing knowledge gap in this area [28].

In a recently published Phase 3 trial, the PELYCAN Trial, the utilization of a peritoneal flap demonstrated reductions in symptomatic lymphoceles (SLCs) from 9.1% to 3.7% (*p* = 0.005), and asymptomatic lymphoceles (ALCs) from 27.2% to 10.3% (*p* < 0.001) over a 6-month follow-up period. Their approach to peritoneal flap creation involved bilaterally incising the ventral peritoneum and fixing it to the pelvic floor. However, it is crucial to highlight that the intervention group experienced a significantly longer operative time, with a duration extended by 11 min compared to the control group (*p* < 0.001) [29].

One of the key benefits associated with employing our technique (PBFB) is the minimal added time required to create the flap (median of 3:15 min). Moreover, the procedure is considered to have a gentle learning curve due to a surgeon’s pre-existing proficiency in anastomosis and suturing. Notably, the areas of the peritoneal bladder that are involved in the sutures lack vascular pedicles and there is no need to incise the peritoneum, significantly reducing the potential for vascular complications or other adverse incidents. Additionally, the financial impact of our technique is negligible as it only needs a single suture. Furthermore, our accumulated experience demonstrates a significant reduction in the median size of lymphoceles. Finally, a short-term follow-up conducted after performing over 550 prostatectomies revealed the absence of symptomatic lymphoceles among these patients. This outcome has subsequently led to a marked reduction in the need for interventional radiology drainage procedures, hospitalizations, and surgical interventions.

Our study is not devoid of limitations, mainly due to its retrospective design and its inherent risk of bias. In addition, this study was conducted in a single center by a single surgeon; therefore, the applicability of our results to other centers may be limited. Furthermore, our study also reported a low lymph node count in both groups, which can be attributed to our adherence to the standard template PLND, as outlined in our technique. However, the absence of a learning curve for such a technique should make it easily reproducible by other surgeons. Furthermore, we also lack the data on long-term data outcomes, including lymphocele development after 3 months (despite the literature stating that most lymphoceles form within 90 days [26]) and post-surgical continence after performing this technique. Nevertheless, it is worth emphasizing that this study introduced a novel technique for peritoneal flap with initially promising results regarding lymphocele size and symptomatic lymphocele development.

## 5. Conclusions

We have demonstrated a modified technique of peritoneal flap eversion (bunching technique) with an initial decrease in postoperative symptomatic lymphoceles and promising short-term outcomes in patients undergoing lymph node dissection in robot-assisted radical prostatectomy. In our experience, this technique does not require a learning curve, and patients had better outcomes at a follow-up after 3 months. While we continue to evaluate long-term outcomes, this preliminary study suggests that the bladder bunching technique is both feasible and safe, without adding significant morbidity or operative time.

## Figures and Tables

**Figure 1 cancers-16-01547-f001:**
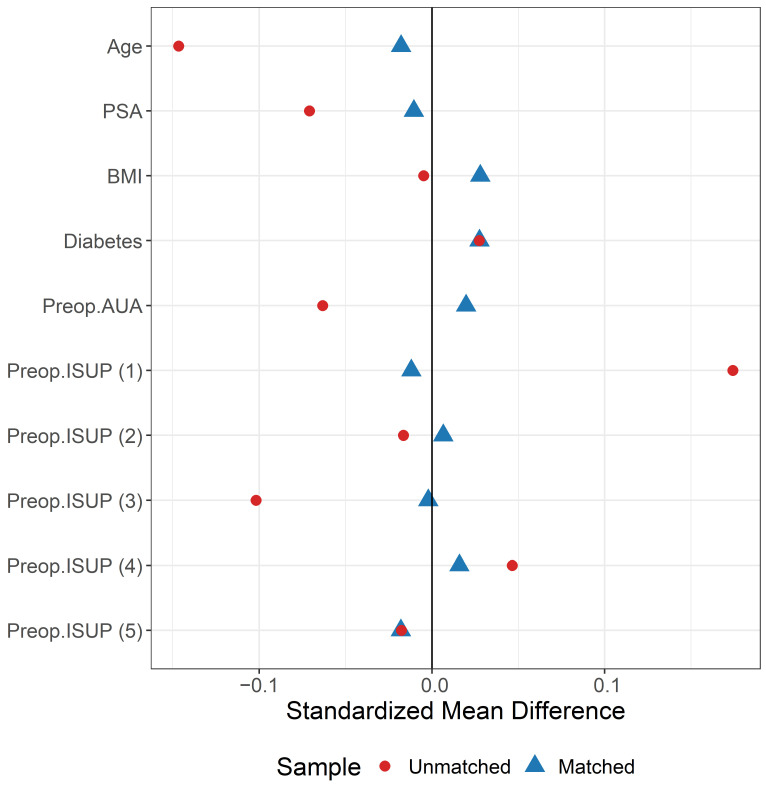
Love plot showing the distributional balance of preoperative variables before (in red) and after (in blue) propensity score (PS) matching.

**Figure 2 cancers-16-01547-f002:**
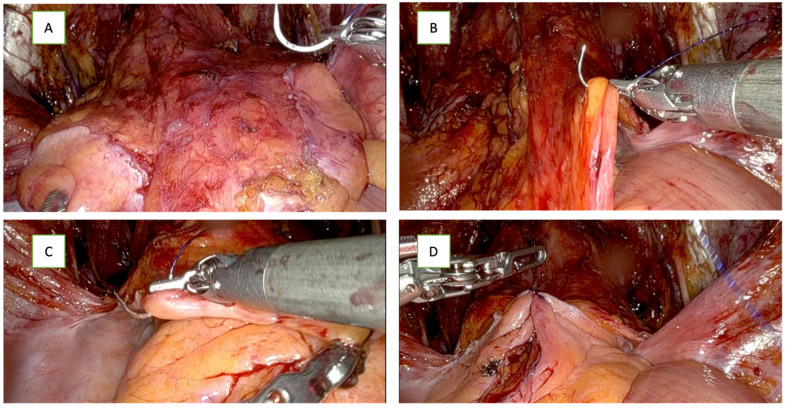
Illustration of the peritoneal flap (bunching technique) after RARP with PLND. (**A**) Peritoneal folds before starting the flap creation; (**B**) first stitch in the peritoneal fold on the right side, medial to the vase; (**C**) stitch of the peritoneal fold on the left side; (**D**) approximating the two edges of the peritoneum in the midline. (**E**) Running sutures in the middle from both sides; (**F**) shape of peritoneal flap after bunching before fixation; (**G**) fixation of the flap to the pubic bone at symphysis pubis; (**H**) final picture of the flap fixed with the lymphadenectomy bed widely opened on both sides.

**Table 1 cancers-16-01547-t001:** Comparison of preoperative patient characteristics (flap vs. control) after 1:3 propensity score (PS) matching, reporting the median value with the interquartile range (IQR) and the number of patients with the percentage; the standardized mean difference evaluates the degree of covariate imbalance. PSA (prostate-specific antigen), BMI (body mass index), diabetes mellitus (DM), Charlson comorbidity index (CCI), AUA (American Urological Association), and ISUP (International Society of Urological Pathology).

Parameters	Flap (*n* = 567)	Control (*n* = 1700)	*p*	Standardized Mean Difference after 1:3 PS Matching
Age (years)	65 (60–70)	65 (60–70)	1.0	−0.18
PSA (ng/mL)	6.4 (4.8–9.6)	6.5 (4.8–9.5)	0.9	−0.015
BMI (Kg/m^2^)	27.8 (25.4–30.9)	28.1 (25.5–30.9)	0.8	0.028
Diabetes (*n*, %)			0.6	
No	479 (84.5)	1419 (83.5)		−0.028
Yes	88 (15.5)	281(16.5)		0.028
Charlson Comorbidity Index (*n*, %)			0.16	
0	14 (2.5)	26 (1.4)		
1–2	335 (59)	963 (56.7)		
3–4	200 (35.3)	635 (37.4)		
>4	18 (3.2)	76 (4.5)		
Preoperative AUA	8	8	1	0.02
	(4–15)	(4–15)		
Biopsy ISUP Grade (*n*, %)			0.9	
Group 1	23 (4.1)	65 (3.8)		–0.012
Group 2	214 (37.7)	647 (38.1)		0.007
Group 3	166 (29.3)	496 (29.2)		0.002
Group 4	96 (16.9)	298 (17.5)		–0.016
Group 5	68 (12)	194 (11.4)		–0.018

**Table 2 cancers-16-01547-t002:** Comparison of preoperative patient characteristics (flap vs. control) before matching, reporting the median value with the interquartile range (IQR) and the number of patients with the percentage; standardized mean difference evaluates the degree of covariate imbalance. PSA (prostate-specific antigen), BMI (body mass index), diabetes mellitus (DM), Charlson comorbidity index (CCI), AUA (American Urological Association), and ISUP (International Society of Urological Pathology).

Parameters	Flap (*n* = 567)	Control (*n* = 4070)	*p*
Age (years)	65 (60–70)	65 (59–69)	0.01
PSA (ng/mL)	6.4 (4.8–9.6)	6.3 (4.6–9.3)	0.2
BMI (Kg/m^2^)	27.8 (25.4–30.9)	27.9 (25.5–30.8)	0.8
Diabetes (*n*, %)			0.5
No	479 (84.5)	3400 (83.5)	
Yes	88 (15.5)	670(16.5)	
Charlson Comorbidity Index (*n*, %)			0.9
0	14 (2.5)	107 (2.6)	
1–2	335 (59)	2394 (58.8)	
3–4	200 (35.3)	1407 (34.6)	
>4	18 (3.2)	162 (4)	
Preoperative AUA	8	8	0.1
	(4–15)	(4–14)	
Biopsy ISUP Grade (*n*, %)			0.002
Group 1	23 (4.1)	335 (8.2)	
Group 2	214 (37.7)	1502 (36.9)	
Group 3	166 (29.3)	1022 (25.1)	
Group 4	96 (16.9)	747 (18.4)	
Group 5	68 (12)	464 (11.4)	

**Table 3 cancers-16-01547-t003:** Comparison of perioperative characteristics in a 1:3 propensity score-matched cohort. Reporting the median value and interquartile range (IQR) and the number of patients with percent total for categorical variables. EBL—estimated blood loss; number of lymph nodes—LN. The *p*-values reported are based on the Wilcoxon two-sample rank-sum test.

Parameters	Flap (*n* = 567)	Control (*n* = 1700)	*p*
EBL (mL)	100 (100–200)	100 (100–150)	<0.001 <0.001
EBL < 100 mL (*n*, %)	323 (57)	1232 (72.5)	
EBL ≥ 100 mL (*n*, %)	244 (43)	468 (27.5)	
Console Time (min)	80 (80–90)	80 (75–90)	<0.001 <0.001
<80 (*n*, %)	58 (10.3)	495 (29.3)	
80–89	275 (48.7)	642 (38)	
≥90	232 (41)	552 (32.7)	
Hospitalization (*n*, %)			0.5
≤1 day	548 (96.7)	1631 (95.9)	
>1 day	19 (3.3)	69 (4.1)	
Postoperative Complications			0.1
(Clavien–Dindo) (*n*, %)			
2	22 (3.9)	27 (1.5)	
≥3	20 (3.5)	48 (2.8)	
Pathological Stage (*n*, %)			0.7s71
pT2	267 (47.2)	815 (47.9)	
≥pT3a	299 (52.8)	885 (52.1)	
Total Number of LN	3 (2–5)	3 (2–5)	0.001
Total Number of LN (*n*, %)			0.001
0	0	19 (1.1)	
1–3	328 (57.9)	832 (50)	
>3	239 (42.1)	849 (49.9)	
Post-op Lymphocele (*n*, %)			
No	431 (76)	1344 (79.1)	0.14
Yes	136 (24)	356 (20.9)	
≤5 cm	124 (21.9)	167 (9.8)	
>5 cm	12 (2.1)	189 (11.1)	0.001
Symptomatic Lymphocele (*n*, %)			0.001
No	567 (100)	1663 (97.8)	
Yes	0 (0)	37 (2.2)	

## Data Availability

The authors confirm that the data supporting the findings of this study are available within the article. Raw data that support the findings are available from the corresponding author (A.G.) upon reasonable request.

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
