# Peer review of "Peritoneal Flap Following Lymph Node Dissection in Robotic Radical Prostatectomy: A Novel “Bunching” Technique"

_cancers, 2024, doi:10.3390/cancers16081547_

Round 1

Reviewer 1 Report (New Reviewer)

Comments and Suggestions for Authors

see remarks in the attachment...low LN-count en low significant lymphoceles in both groups are strong limitations in this study and makes it hard to draw any conclusions.

nonetheless, none of the above is mentioned in the discussion section

Author Response

Thanks for reviewing our article and including substantial comments to further improve the quality of our study. We have carefully addressed each of your comments and concerns in RED and we have highlighted all changes in the manuscript using the strikethrough feature.

I attached Point to Point response 

Reviewer 2 Report (New Reviewer)

Comments and Suggestions for Authors

Dear authors, I read with interest your manuscript. The text is well written and research well conducted methodologically. However, the concept behind this study is not innovative and the benefit of peritoneal flap, or peritoneal reconstruction aimed to divide LND area (obturators fossa) with the retropubic space was already described (Robot-Assisted Extended Pelvic Lymph Nodes Dissection for Prostate Cancer: Personal Surgical Technique and Outcomes. Int Braz J Urol. 2015 Nov-Dec;41(6):1209-12019. doi: 10.1590/S1677-5538.IBJU.2015.0055. PMID: 26742982; PMCID: PMC4756950.) Please debate about this concept into the discussion.

Furtheromore, after your descried reconstruction, the completely cover the anastomosis, what about postoperative bleeding? I mean, if a bleeding occurred with hematoma, considering that the spase close to the anastomosis is closed by the suture, is there the risk of an excessive tension at the level of the anastomosis?

Author Response

Thanks for reviewing our article and including substantial comments to further improve the quality of our study. We have carefully addressed each of your comments and concerns in RED and we have highlighted all changes in the manuscript using the strikethrough feature.

I attached Point to Point response 

Reviewer 3 Report (New Reviewer)

Comments and Suggestions for Authors

The author highlighted the novel technique for preventing lymphocele after prostatectomy with PLND. The reviewer agrees with some of the content; however, the reviewer has proposed several suggestions to enhance the paper:

Major point

1.     The authors should indicate the reason for distinguishing between 5cm of lymphocele. 

2.     The authors should show the range of PLND for NCCN low risk group, such as limited/standard.

3.     The authors should indicate PLND time, the presence of nerve sparing and pT stage/resection margin.

4.     The reviewer recommends the multivariate analysis for the symptomatic lymphocele.

5.     The authors should show the method for screening the asymptomatic lymphocele.

6.     The authors should discuss the adaptation for the case with extended PLND.

7.     The authors should discuss the complication caused by peritoneal fenestration.

Minor point

1.     What kind of the suture do you use with PBFB.

2.     The author should revise some double space in this article.

Author Response

Thanks for reviewing our article and including substantial comments to further improve the quality of our study. We have carefully addressed each of your comments and concerns in RED and we have highlighted all changes in the manuscript using the strikethrough feature.

I attached Point to Point response 

Round 2

Reviewer 1 Report (New Reviewer)

Comments and Suggestions for Authors

Coul agree to this study now, with these adjustments. Nonetheless, no statement is given to the low amount of LN's per patient.

in over 50% of pts in both groeps have 3 or less LN's in there pathology report. I do think international guidelines recommend a median count of 10.

this support the idea that the PLND was limited en therefore no high percentage of lymfoceles were seen in both groups.

maybe I can't find it, but i did not read this anywhere in the discussion

Author Response

Coul agree to this study now, with these adjustments. Nonetheless, no statement is given to the low amount of LN's per patient.

in over 50% of pts in both groeps have 3 or less LN's in there pathology report. I do think international guidelines recommend a median count of 10.

this support the idea that the PLND was limited en therefore no high percentage of lymfoceles were seen in both groups.

maybe I can't find it, but i did not read this anywhere

Thank you for your notice, we added and highlighted the following part to the last paragraph in the discussion under the limitation to our study

(Furthermore, our study also reported a low lymph node count in both groups, which can be attributed to our adherence to the standard template PLND as outlined in our technique.)

Once again, we would like to thank you for reviewing our article and including substantial comments to further improve the quality of our study.

This manuscript is a resubmission of an earlier submission. The following is a list of the peer review reports and author responses from that submission.

Round 1

Reviewer 1 Report

Comments and Suggestions for Authors

Thank you for the opportunity to review this manuscript. 

I have several questions and recommendations as below;

  1. The authors cited several references (#14-17) to show the advantage of peritoneal flap. However, technically, #14, #15, and #16 indicated a different technique (peritoneal interposition), which is not used in the present study. Moreover, #17 did not use the peritoneal flap (just tried the transperitoneal approach). Are there any other references that specifically used the “bunching” technique? Or is this a widely accepted surgical technique? Otherwise, the authors should validate their “new” technique first, comparing peritoneal interposition.

  2. What is the definition of “symptomatic lymphocele”? And when did you diagnose it? Because it is the only significant result, you have to focus on it.

  3. Is there any reason that the authors divide the console time into three groups? 

  4. What is the meaning of the “total number of LN”? Does it mean that the number of dissected lymph nodes? Then, you have to discuss it because it may be related to postoperative lymphatic drainage.

  5. The advantage of PBFB seems to be allowing lateral gutters to open and keeping the obturator fossa unobstructed. However, it appears too late in the manuscript to read, so I recommend relocating that part to the earlier part of the manuscript. Also, comparing with the other PIFs.

  6. The authors described that it can reduce the potential for vascular complications because the bladder neck lacks vascular pedicles, but in the results, the EBL is higher in the Flap Group.

  7. According to their conclusion. PBFB demonstrated a decreased postoperative symptomatic lymphocele and promising short-term outcomes. However, there was no discussion about these findings except for referring to previous studies.

  8. Is there any evidence regarding the learning curve? It seems to be inappropriate to refer to a gentle learning curve or easily reproducible without evidence. Undoubtedly, it is the bias and limitation that data obtained from a single surgeon without blindness.

  9. The author described that the financial impact is negligible as it only needs a single suture. Is this a benefit? Do you mean that if a surgeon does choose to use this PBFB, the surgeon may not need to other hemostatic agents like Floseal, Tachosil, etc? Even in some countries other than the USA, the number of surgical sutures may be meaningless because of the bundled payment.

Reviewer 2 Report

Comments and Suggestions for Authors

The authors are to be congratulated on a well-written manuscript.  I have a few questions and suggestions for the study.

In the design and results, my main concern is that there is no timeframe stated for dates of surgery.  It is unclear to me whether or not the flap group and the matched control group had their surgeries during the same timeframe.  I believe this should be examined and stated.  Generally in a retrospective study such as this, the surgeon appropriately modifies his technique after learning that it may improve surgical outcomes.  He then carefully examined his results to see if his patients have benefited.  Therefore, the experimental group tends to be more recent than the control group.  I do think this needs to be examined here.

Along these lines, I am curious why the control group all underwent pelvic ultrasounds.  Is this standard of care for the surgeon?  If so, why?

Finally, I feel there are several randomized prospective studies along with meta analyses that are not cited in the discussion and should be.  Certainly the recently published PELYCAN trial should be included as well as PreFix and PLUS.  I also feel the meta analysis published by May (published prior to PELYCAN and including PIANOFORTE, PerFix, ProLy, and PLUS) should be cited.